# *Hyperion*: Fused Multi-trial and Gradient Descent for Joint Hyperparameter and Neural Architecture Optimization

## Abstract

We consider the fusion of multi-trial optimizers and gradient descent based one-shot algorithms to jointly optimize neural network hyperparameters and architectures. To combine strengths of optimizers from both categories, we propose *Hyperion*, which smartly distributes searched parameters into different involved optimizers, efficiently samples sub-search-spaces to reduce exploration costs of one-shot algorithms, and orchestrates co-optimization of both hyperparameters and network architectures. We demonstrate with open and industrial datasets that *Hyperion* outperforms non-fused optimization algorithms in optimized metrics, while significantly reducing GPU resources required for one-shot algorithms.

## 1 Introduction

Hyperparameter optimization (HPO) (Bischl et al., 2023) and neural architecture search (NAS) (White et al., 2023) are two important and often separately performed tasks for AutoML. Typically, one first selects hyperparameters for data pre-processing options, feature engineering methods and training configurations. As a second step, neural architecture search is triggered to construct a corresponding machine learning pipeline, perform training and identify best performing network architectures. After those steps, hyperparameters can be further tuned for best performing architectures.

The separation of HPO and NAS could lead to sub-optimal results, as best performing architectures tend to depend on the selected hyperparameters (He et al., 2021). To automate hyperparameter and neural architecture selection and jointly optimize them together, one could extend common multi-trial or one-shot methods (White et al., 2023; He et al., 2021) for neural architecture search to cover additionally the search space of hyperparameters. Indeed, the underlying optimization algorithms adopted for NAS like random search, bayesian optimization, reinforcement learning, etc. are already applied alone to hyperparameter optimization (Bischl et al., 2023; Falkner et al., 2018b).

Extending NAS algorithms to co-optimize hyperparameters, however, is not straightforward to be done in an efficient and effective manner. Multi-trial algorithms (Suganuma et al., 2017; Zoph & Le, 2017; Li & Talwalkar, 2019), while being typically generic, assume training from scratch in each trial for new hyperparameters and network architectures. This could introduce inefficiency into optimization as learnt network parameters are discarded across different trials. Gradient descent based one-shot algorithms (Liu et al., 2019; Chen et al., 2021; Cai et al., 2019), on the other hand, sample new architectures from a shared supernet and optimize network weights and architectures simultaneously in the training loop. This could greatly help to improve optimization as (1) network weights are kept and continuously explored for different model architectures and (2) network architectures are optimized following calculated gradients. However, one-shot algorithms can require more GPU resources to train the super network. Moreover, it is non-trivial or infeasible to incorporate many conventional hyperparameters like data cleaning methods, learning rate, batch size, etc., into gradient descent based one-shot algorithms due to the difficulty to backpropagate gradients to these hyperparameters.

We present in this paper *Hyperion* to perform joint hyperparameter and neural architecture search in an efficient manner. *Hyperion* achieves this by fusing multi-trial and gradient descent based one-shot algorithms together to cover a large search space while intelligently reducing GPU resources

needed. Our detailed contributions are as follows: *(i)* We introduce a smart splitter – a learning-based algorithm that automatically finds distributions of searched parameters to multi-trial and one-shot algorithms to search for best performing neural network models. *(ii)* We design a smart sub-search-space sampler that automatically learns the best sub-search-spaces for one-shot algorithms to reduce their GPU resource usage. *(iii)* We integrate the smart splitter, smart sub-search-space sampler, multi-trial optimizers like TPE (Bergstra et al., 2011), Anneal (Fischetti & Stringher, 2019) and BlendSearch (Wang et al., 2021), and the popular gradient descent based one-shot algorithm DARTS into *Hyperion* to orchestrate co-optimization of hyperparameters and neural architectures. We demonstrate with extensive experiments that *Hyperion* outperforms standalone non-fused multi-trial and one-shot algorithms in terms of optimization metrics, while at the same time significantly reducing GPU resource utilization.

## 2 RELATED WORK AND BACKGROUND

### 2.1 HYPERPARAMETER OPTIMIZATION AND NEURAL ARCHITECTURE SEARCH

To set up a machine learning pipeline, many parameters need to be configured besides network architecture parameters, e.g data pre-processing options like data cleaning (Chu et al., 2016) and augmentation (Mikołajczyk & Grochowski, 2018), feature engineering techniques (Zheng & Casari, 2018) including selection, construction and extraction of features and training related hyperparameters (Feurer & Hutter, 2019) like batch size, learning rate schedule, stochastic gradient decent optimizers, etc. Well known optimization techniques like random search (RS), bayesian optimization (BO) (Shahriari et al., 2015), evolutionary algorithms (EA) (Bäck & Schwefel, 1993), reinforcement learning (RL) (Sutton & Barto, 2018), etc., have already been successfully applied here.

Most neural architecture search (NAS) methods assume fixed hyperparameters during their search processes (White et al., 2023; He et al., 2021). In terms of underlying optimization algorithms, we see indeed many popular algorithms adopted for hyperparameter optimization are also applied to NAS, e.g. EA, RL and BO (He et al., 2021). For sampling neural network architectures, state-of-the-art techniques differ in whether they sample a new network architecture and train it from scratch or train a common super network and consequently sample and reuse the weights from this super network (White et al., 2023). In the former case, multiple trials of complete training of different network architectures have to be performed, giving rise to the name multi-trial. In the latter case, architecture search and training are tightly coupled or entirely unified; due to this, they are often termed as one-shot algorithms. We focus on gradient descent based one-shot algorithms like DARTS (Liu et al., 2019), P-DARTS (Chen et al., 2021) and ProxylessNAS (Cai et al., 2019), which unify architecture searching and weight update in the same training loop, further enhancing NAS.

Joint optimization of hyperparameters and neural architectures remains little explored even though the best network architecture could depend on the chosen machine learning pipeline hyperparameters. AutoHAS (Dong et al., 2020) proposed to extend reinforcement learning to not only sample architecture parameters but also hyperparameters. NARS (Dai et al., 2021) trained a single predictor to jointly score architecture and hyperparameters. Extensions of several existing multi-trial techniques for joint HPO and NAS were proposed in (Guerrero-Viu et al., 2021; Zela et al., 2018). To our best knowledge, *Hyperion* is the first to propose a generic framework fusing multi-trial and one-shot optimizers in joint HPO and NAS, while addressing their respective shortcomings.

### 2.2 DIFFERENTIABLE NEURAL ARCHITECTURE SEARCH

DARTS incorporates architecture search into the training process of neural networks by creating a super network (supernet) with edges that can be turned "on and off". Essentially, between a pair of nodes (tensors), there can be multiple parallel edges, each specifying a different operation (e.g. combination of different operators, kernel size, activation function, etc.). To sample edges from this network and create new architectures, each edge is additionally assigned a weight, such that the parallel operations output a weighted sum of individual operations. By backpropagating loss to architecture weights, their contributions to the final model accuracy can be determined and the best operation/edge can be selected. Formally, assuming parallel edges from node $i$ to $j$, the averaged operation for all those operations applied to tensor $x^{(i)}$, $\bar{o}^{(i,j)}(x^{(i)})$, becomes

Figure 1: An example super network in DARTS with 3 nodes for each cell, 3 candidate operations between any pair of nodes (red, green and blue) and a depth of 4 stacked cells. In each cell, $I_1$ and $I_2$ are the two input nodes created by copying the input in two branches. $O$ is the output of the cell that is the concatenation of the output of all nodes (excluding input nodes). After training, only edges with top weights are kept to sample a final network, e.g. in the original DARTS implementation, 2 incoming edges with the highest weights are kept for each node.

$$\bar{o}^{(i,j)}(x^{(i)}) = \sum_{o \in O} \frac{exp(\alpha_o^{(i,j)})}{\sum_{o' \in O} exp(\alpha_{o'}^{(i,j)})} o(x^{(i)}) \tag{1}$$

Here, for each operator $o \in O$, there is an associated architecture weight $\alpha_o^{(i,j)}$. Note that, gradients are backpropagated to both architecture and operator weights in training and, in the end, the best network architecture can be selected by keeping the edges with highest weights, i.e. $o^{(i,j)} = \arg\max_{o \in O} \alpha_o^{(i,j)}$. Note that, DARTS-based algorithms do not construct a search space for the entire network, but rather on the cell level, see Figure 1. This is inspired by the fact that most successful networks like ResNet (He et al., 2016) and Transformer (Vaswani et al., 2017) are composed by stacking identical layers. In practice, this also helps to reduce search complexity by focusing on a small proven search space.

## 3 Hyperion

In this section, we first present the challenges in fusing multi-trial and one-shot optimizers. This is followed by an in-depth explanation of each learning-based algorithm involved in *Hyperion* and how they jointly solve the challenges in fusion.

### 3.1 CHALLENGES AND SOLUTION OVERVIEW

Although with the fusion of multi-trial and one-shot algorithms we could potentially combine their strengths while circumventing individual shortcomings, a few important intertwined challenges arise during this process:

1. **Distribution:** In fusion, neural architecture related parameters can be optimized by either type of optimizers. It is not straightforward to distribute those parameters to the involved optimizers as (1) it is unclear which type of optimizers could yield the best final model accuracy (Li & Talwalkar, 2019) and (2) there exists a non-trivial trade-off between optimization efficiency and resource usage for the two types of optimizers. It is therefore critical to distribute the hyperparameters between them while addressing those concerns.

2. **Computational Resource:** For performing neural architecture search, DARTS would require huge GPU memory as a supernet is trained to incorporate all possible network architectures. Therefore, it is necessary to reduce the required resource usage while ensuring best performing network architectures can be still found.

3. **Orchestration:** One last challenge we would like to highlight is the orchestration of multi-trial and one-shot algorithms, as in fusion, they need to work together to create and optimize the supernet and search for best performing architectures.

We present in this paper a generic framework *Hyperion* to address the above-mentioned challenges, as shown in Figure 2. We have as input the collected dataset and the search space covering both hyperparameters and network architectures, specified by a configuration file. Additionally, this file contains various settings for the experiment, such as time budget for searching, optimizer used when training, optimization objectives, etc.

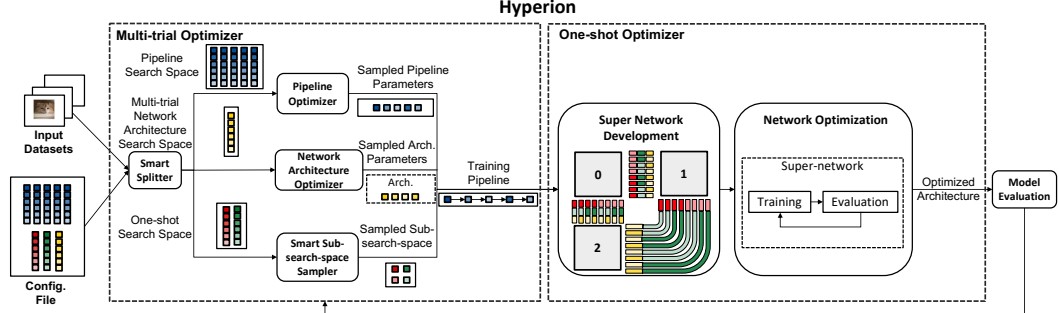

Figure 2: Overview of *Hyperion*: Different searched parameters are shown with different stacks of squares. The number of choices for each parameter is indicated by the number of squares in the stack (color coded with different brightness to reflect various choices). Blue-colored stacks refer to ML pipeline hyperparameters, such as the number of epochs, learning rate, etc., that the one-shot optimizer cannot optimize. Red, green and yellow colored stacks refer to parameters that can be optimized by both multi-trial and one-shot optimizers, such as kernel size, layer operation, etc.

In *Hyperion*, one can view the multi-trial optimizer as the outer optimization loop, and the one-shot optimizer as the inner optimization loop. First, the entire search space is processed by the learning-based smart splitter, which distributes parameters between the outer and inner optimization loops. In Figure 2, all blue-colored stacks (searched parameters) are distributed to the multi-trial optimizer, which samples values for hyperparameters that the one-shot optimizer cannot optimize. Additionally, there are searched parameters, which the inner optimization loop can optimize, but nevertheless are assigned to the outer loop optimizer by the smart splitter (color-coded in yellow). After distribution, the learning-based sub-search-space sampler extracts a sub-search-space from the original search space assigned to the one-shot optimizer, in an attempt to reduce resource usage required in the inner optimization loop. Subsequently, a machine learning pipeline is constructed according to selected hyperparameters and a supernet is created with configured parameters by the multi-trial optimizer and a sampled sub-search-space. This supernet is trained using the constructed training pipeline by the one-shot optimizer to simultaneously search model weights and architecture. Final evaluation results of the optimized model are fed back to the smart splitter, smart sub-search-space sampler, and multi-trial pipeline and network architecture optimizers to learn better parameter distribution, sub-search-space sampling and network and pipeline choices for subsequent trials.

## 3.2 MULTI-TRIAL OPTIMIZERS IN *Hyperion*

Multi-trial optimizers in *Hyperion* are responsible to (1) distribute searched parameters between multi-trial and one-shot optimizers, (2) sample machine learning pipeline related hyperparameters (data cleaning, feature engineering, training setups like number of epochs and learning schedule, etc.), (3) optimize architecture related parameters not covered by the one-shot optimizer and (4) automatically reduce the search space for the one-shot optimizer.

**Smart Splitter.** To solve the distribution challenge and balance between optimization efficiency and resource usage, we adopt a smart splitter as also shown in Figure 2. We create a search space for smart splitter by assigning a binary variable for each parameter in our search space $\Theta$ to indicate whether it is distributed to the multi-trial or one-shot optimizer: $\Theta_{split} = \{sp_\theta \in \{0, 1\} \mid \theta \in \Theta\}$. Invalid distributions are simply discarded. One challenge for splitting is to divide the architecture search among multi-trial and one-shot optimizers. A simple way would be to assign a subset of parallel edges among nodes (see Figure 1) to the multi-trial optimizer and the rest to the one-shot optimizer. However, this would mean many potential searched architecture parameters for the multi-trial optimizer with large networks. We instead do a splitting on a coarse granularity by assigning high-level architecture parameters like kernel size and operator type. This helps to simplify the design space for multi-trial optimizers while still allowing complete cell structure exploration.

**Pipeline and Multi-trial Architecture Optimizers.** After splitting, both machine learning pipeline related parameters and a subset of neural architecture related parameters are assigned to the outer optimization loop with multi-trial optimizers. The pipeline search space $\Theta_{pipeline}$ can be straight-forwardly constructed for involved categorical (e.g. data cleaning methods) or numerical parameters (e.g. weight decay) by joining their individual search spaces together.

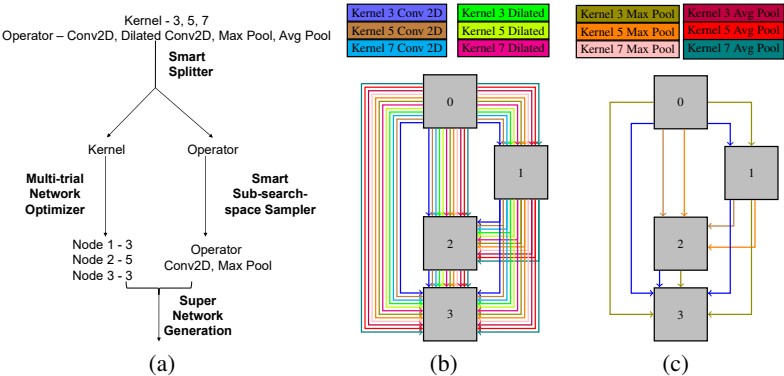

Figure 3: Cell construction for a super network in *Hyperion*. Example shows 3 kernels {3,5,7} and 4 layer operations {Conv2D, Dilated Conv2D, Max Pool, Avg Pool} as architecture parameters. (a) In *Hyperion*, the search space is divided between multi-trial and one-shot optimizers, with network architecture parameters jointly searched and reduced. (b) Original DARTS cell structure is composed using all available parameter combinations. (c) In *Hyperion*, a reduced cell structure is constructed using proposed parameters by both the multi-trial network optimizer and the smart sub-search-space sampler, as shown in (a).

The architecture optimization at this stage is a bit more involved. As discussed, the smart splitter assigns high-level architecture parameters like kernel size and activation function to the optimizers. Given such information, for performing architecture optimization during multi-trial optimization, we search these parameters for each node in the basic cell structure, see Figure 3. This means that for each node, the multi-trial optimizer will individually configure the assigned parameter like operation type and activation function. Equivalently, this helps to reduce the number of parallel edges between each pair of nodes by offloading the search of certain high-level architecture parameters to the outer optimization loop. In the example, kernel sizes for each node are directly configured by the multi-trial optimizer. After training the super network in each trial, the final network can be sampled using architecture weights in the super network. In the corner case when all architecture parameters are assigned to the multi-trial optimizer, sampling a final cell structure is exclusively done at this stage as there is no involved one-shot optimizer.

**Smart Sub-search-space Sampler.** The smart splitter assigns a subset of architecture search spaces to the one-shot optimizer DARTS. Nevertheless, for large number of parallel edges and nodes in a cell, it would require significant GPU memory for training the super network. To further improve resource efficiency of DARTS, we use a sub-search-space sampler to extract a subset of design choices available for those parameters distributed to the one-shot optimizer. For our example in Figure 3, it samples operators {Conv2D, Max Pool} from the original search space {Conv2D, Dilated Conv2D, Max Pool, Avg Pool}, which is then used for constructing a super network with reduced size in *Hyperion*. This reduced search space for the one-shot optimizer should still contain design choices that can produce the best results. However, determining this sub-search-space is non-trivial as there are in general no clear rules to follow. Therefore, multi-trial optimization algorithms (e.g. reinforcement learning or Bayesian optimization based), which on the fly learn the optimized sub-search-space based on past trials, are preferred. Technically, we encode for each parameter its possible values with a multi-hot vector: a value 1 in this vector means that the corresponding value is considered in the one-shot optimizer. Our proposed sub-search-space sampler learns automatically with progressing trials best choices for those multi-hot vectors.

### 3.3 ONE-SHOT OPTIMIZER IN *Hyperion*

The *Hyperion* framework shown in Figure 2 is generic and can work with different one-shot optimizers. We focus in this paper on DARTS as our chosen one-shot optimizer, the family of which would be regarded as most effective among NAS techniques due to its capability to optimize model architecture and operator weights simultaneously (White et al., 2023).

To integrate DARTS in *Hyperion*, the main task is to construct a modified super network in tandem with the multi-trial optimizers, as they share the responsibility for optimizing network architectures. We thus extended DARTS to construct a supernet using outputs of smart splitter, smart sub-search-

---

**Algorithm 1** *Hyperion* for joint hyperparameter and neural architecture optimization

---

**Input:** dataset $D$, target function $F$, search space $\Theta$, allowed number of trials $T$, number of trials $n_{init}$ for initialization

**Output:** Optimized ML configurations $\boldsymbol{\theta}^* = [\boldsymbol{\theta}^*_{split}, \boldsymbol{\theta}^*_{pipeline}, \boldsymbol{\theta}^*_{subspace}, \boldsymbol{\theta}^*_{arch\_m}, \boldsymbol{\theta}^*_{arch\_o}]$

**for** $t = 1$ **to** $n_{init}$ **do**                                                                                  ▷ Initial Exploration
 generate random split $\boldsymbol{\theta}_{(split,t)}$
 $\Theta_{pipeline}, \Theta_{arch\_m}, \Theta_{subspace}, \Theta_{arch\_o} \leftarrow GetSearchSpaces(\Theta, \boldsymbol{\theta}_{(split,t)})$
 initialize randomly $\boldsymbol{\theta}_{x,t}, \forall x \in \{pipeline, arch\_m, subspace\}$
 **if** $\Theta_{arch\_o,t} \neq \emptyset$ **then**
  $S \leftarrow GenerateModifiedSuperNetwork(\boldsymbol{\theta}_{(arch\_m,t)}, \boldsymbol{\theta}_{(subspace,t)}, \Theta_{arch\_o})$
  **while** *not converged* **do**                                                                     ▷ One-shot Optimizer
   perform gradient descent to update $\boldsymbol{\alpha}$ and $\boldsymbol{w}$ of supernet $S$
 $e_t \leftarrow F(D, [\boldsymbol{\theta}_{(pipeline,t)}, \boldsymbol{\theta}_{(arch\_m,t)}, \boldsymbol{\theta}_{(arch\_o,t)}])$                                 ▷ Model Evaluation
 $\mathbf{H}_x \leftarrow \mathbf{H}_x \cup (\boldsymbol{\theta}_{(x,t)}, e_t), \forall x \in \{split, pipeline, arch\_m, subspace\}$
**for** $t = n_{init} + 1$ **to** $T$ **do**                                                               ▷ Exploration + Exploitation
 $M_{split} \leftarrow FitModel(\mathbf{H}_{split})$                                               ▷ Smart Splitter Multi-trial Optimizer
 $\boldsymbol{\theta}_{(split,t)} \leftarrow \arg\max_{\boldsymbol{\theta}_{split} \in \Theta_{split}} A(\boldsymbol{\theta}_{split}, M_{split})$
 $\Theta_{pipeline}, \Theta_{arch\_m}, \Theta_{subspace}, \Theta_{arch\_o} \leftarrow GetSearchSpaces(\Theta, \boldsymbol{\theta}_{(split,t)})$
 $M_{pipeline} \leftarrow FitModel(\mathbf{H}_{pipeline})$                                                         ▷ Multi-trial Optimizer
 $\boldsymbol{\theta}_{(pipeline,t)} \leftarrow \arg\max_{\boldsymbol{\theta}_{pipeline} \in \Theta_{pipeline}} A(\boldsymbol{\theta}_{pipeline}, M_{pipeline})$
 $M_{arch\_m} \leftarrow FitModel(\mathbf{H}_{arch\_m})$                                                           ▷ Multi-trial Optimizer
 $\boldsymbol{\theta}_{(arch\_m,t)} \leftarrow \arg\max_{\boldsymbol{\theta}_{arch\_m} \in \Theta_{arch\_m}} A(\boldsymbol{\theta}_{arch}, M_{arch})$
 $M_{subspace} \leftarrow FitModel(\mathbf{H}_{subspace})$                                  ▷ Smart Sub-search-space Sampler Multi-trial
 $\boldsymbol{\theta}_{(subspace,t)} \leftarrow \arg\max_{\boldsymbol{\theta}_{subspace} \in \Theta_{subspace}} A(\boldsymbol{\theta}_{subspace}, M_{subspace})$
 **if** $\Theta_{arch\_o,t} \neq \emptyset$ **then**
  $S \leftarrow GenerateModifiedSuperNetwork(\boldsymbol{\theta}_{(arch\_m,t)}, \boldsymbol{\theta}_{(subspace,t)}, \Theta_{arch\_o})$
  **while** *not converged* **do**                                                                     ▷ One-shot Optimizer
   perform gradient descent to update $\boldsymbol{\alpha}$ and $\boldsymbol{w}$ of supernet $S$
 $e_t \leftarrow F(D, [\boldsymbol{\theta}_{(pipeline,t)}, \boldsymbol{\theta}_{(arch\_m,t)}, \boldsymbol{\theta}_{(arch\_o,t)}])$                                 ▷ Model Evaluation
 $\mathbf{H}_x \leftarrow \mathbf{H}_x \cup (\boldsymbol{\theta}_{(x,t)}, e_t), \forall x \in \{split, pipeline, arch\_m, subspace\}$
$i^* \leftarrow \arg\min_i \{e_i\}$                                                               ▷ $\{e_i\}$ is the set of all evaluations
**return** $[\boldsymbol{\theta}_{(split,i^*)}, \boldsymbol{\theta}_{(pipeline,i^*)}, \boldsymbol{\theta}_{(subspace,i^*)}, \boldsymbol{\theta}_{(arch\_m,i^*)}, \boldsymbol{\theta}_{(arch\_o,i^*)}]$

---

space sampler, and network optimizer in the outer optimization loop. Taking again our running example in Figure 3, between each pair of nodes, the available parallel edges are determined by the chosen kernel sizes for each node and the reduced search space on operator. For example, for node 1, only Conv2D and Max Pool with kernel size 3 are available. By traversing all nodes and keeping edges for available combinations of architecture parameters in the reduced cell search space, *Hyperion*'s DARTS cell structure is constructed. The super network is then constructed by repeatively stacking those cells, as shown in Figure 1. Finally, the original DARTS training and architecture optimization are performed on this supernet.

*Hyperion* intelligently engages one-shot optimizers and significantly reduces search space for such optimizers. This leads to greatly reduced GPU resource usage and improved overall optimization efficiency. A detailed analysis for this is given in Appendix B.

### 3.4 *Hyperion* ALGORITHM

Formally, we notate the supernet as $S$, its architecture weights as $\boldsymbol{\alpha}$, and model weights as $\boldsymbol{w}$. We use $t$ to index the $t^{th}$ trial as initiated by the outer optimization loop, and in each trial, we adopt a corresponding acquisition function $A$ for the smart splitter, the smart sub-search-space sampler and multi-trial optimizers to acquire new searched configurations: $\boldsymbol{\theta}_{(split,t)}$ as the $t^{th}$ sampled parameter distribution, $\boldsymbol{\theta}_{(pipeline,t)}$ as the $t^{th}$ sampled machine learning pipeline configuration, $\boldsymbol{\theta}_{(arch\_m,t)}$ as the $t^{th}$ sampled neural architecture parameters by the multi-trial optimizer and $\boldsymbol{\theta}_{(subspace,t)}$ as the $t^{th}$ sampled sub-search-space for the inner optimization loop. Accordingly, for the inner optimization loop, $\boldsymbol{\theta}_{(arch\_o,t)}$ is the $t^{th}$ sampled architecture parameter by DARTS. Note again here that in *Hyperion* both multi-trial and one-shot optimizers participate in neural architecture optimization such that $\boldsymbol{\theta}_{(arch\_m,t)}$ and $\boldsymbol{\theta}_{(arch\_o,t)}$ jointly determines the $t^{th}$ best neural architecture. We record with $e_t$ the evaluated result for the $t^{th}$ trial. We use $\mathbf{H}_{split}$ to keep history of all the explored parameter distributions by the smart splitter and their evaluations *i.e.*, $\mathbf{H}_{split} = \{(\boldsymbol{\theta}_{(split,1)}, e_1), \ldots, (\boldsymbol{\theta}_{(split,T)}, e_T)\}$,

which is used to train and update the smart splitter's internal model $M_{split}$ in each trial to improve its performance. Similarly, $\mathbf{H}_{subspace}$, $\mathbf{H}_{pipeline}$ and $\mathbf{H}_{arch\_m}$ keep histories of all the explored configurations by the respective optimizers in the outer optimization loop, which are used to train and improve their internal models ($M_{subspace}$, $M_{pipeline}$ and $M_{arch\_m}$) in each trial for better optimization. With the introduced notations, *Hyperion* framework as shown in Algorithm 1 can be explained as follows:

We initialize the entire search space by randomly generating $n_{init}$ samples. The initialization is required to bootstrap the internal models used by various multi-trial optimizers involved. Selection of $n_{init}$ depends on the search space size. For a larger search space, $n_{init}$ should be also be larger for bootstrapping. If no parameter is distributed to the one-shot optimizer, the multi-trial network optimizer samples directly a final cell structure for the super network. Otherwise, *Hyperion* generates a modified supernet as explained in Section 3.3. This supernet is generated using proposed configurations of the smart splitter, smart sub-search-space sampler, model architecture parameters decided by the multi-trial network optimizer, and search space assigned to the one-shot optimizer as decided by the smart sub-search-space sampler. *Hyperion* optimizes the model architecture and weights in a single training process, similar to the original DARTS. After this, it samples the best model architecture configuration from the super network. *Hyperion* evaluates the final model and stores all parameter configurations and evaluation results in the exploration history.

Once initialized, for subsequent trials, *Hyperion* fits probabilistic models for different involved multi-trial optimizers like the smart splitter and smart sub-search-space sampler on the explored history. Those models help the involved optimizers to quickly suggest what next parameter configurations to search. Learning such models to guide exploration are done differently for the different underling optimizers. For random search, one keeps simply a list of all explored configurations without any model. For machine learning based surrogate models, they fit machine learning models to predict from sampled configurations to final model performance. For Bayesian optimization methods like TPE, they fit a probability distribution function over the history to estimate the performance of new configurations. In multi-trial optimizers based on heuristics like Anneal, specially designed rules are used for extracting new configurations. Based on new proposed parameter configurations, if any is distributed to the one-shot optimizer, then *Hyperion* generates a modified super network. For this step, *Hyperion* requires the extracted configurations in the current trial and the search space of neural architectures. After training the super network, the best model architecture is determined and consequently evaluated and added to the history. Note that in case of multi-objective optimization, all objectives are evaluated and their weighted sum is used by both the outer and inner optimization loops. After all trials are explored, the final best sample from the considered search space and the corresponding model architecture are returned.

## 4 EXPERIMENTS AND RESULTS

In this section we experimentally evaluate *Hyperion* with open image datasets and an industrial use case. After presenting our experiment setup, we first evaluate the effectiveness of *Hyperion* in optimizing neural networks. We then study optimization reproducibility and efficiency of *Hyperion* with open datasets. Lastly, we ablate the design of Hyperion by experimenting with smart splitter and smart sub-search-space sampler.

### 4.1 EXPERIMENTAL SETUP

*Baselines.* We used three different types of multi-trial optimization algorithms - TPE (based on Bayesian optimization), Anneal (based on heuristics), and BlendSearch (based on a combination of global Bayesian optimization and reinforcement learning), and a one-shot NAS algorithm - DARTS. We use a single multi-trial optimizer for the entire outer optimization loop of *Hyperion*.

*Configuration.* For a fair comparison, we ensure that the search space and run time are identical for all algorithms in each experiment. To ensure that the total experiment time is feasible, we optimize parameters which can be optimized by both multi-trial and one-shot optimizers, such as kernel size, activation function, layer operation and model architecture. We fix hyperparameters defining data preprocessing, feature engineering and training. We use the industrial motor dataset to show the impact of joint hyperparameter and neural architecture optimization. Our detailed experiment setup is shown in Appendix C and D. The experiments of CIFAR10 (Krizhevsky, 2009) run for 96 hours each, MNIST (LeCun & Cortes, 2010) run for 72 hours each, Fashion-MNIST (Xiao et al., 2017)

run for 20 hours each, EuroSAT (Helber et al., 2019) run for 20 hours each and the industrial motor dataset run for 20 hours each on a NVIDIA A6000 GPU with 48 GB memory.

## 4.2 OPTIMIZATION EFFECTIVENESS

We apply *Hyperion* and its baseline algorithms to five different datasets/problems, where both the model size and accuracy are jointly optimized with a weighted sum for CIFAR10 (Equation 2) and industrial motor (Equation 3) datasets, and only model accuracy is optimized for MNIST, Fashion-MNIST and EuroSAT datasets. We report the best model found for each algorithm and dataset (details in Appendix E). We observe that for all datasets *Hyperion* outperforms the baseline multi-trial optimizer (BlendSearch) by 1.1% - 19.0%, and it outperforms the baseline one-shot optimizer DARTS by 1.6% - 6.8%. *Hyperion* fuses both types of optimizers together and in comparison to the baseline multi-trial optimizer, it explores more search space during the same experiment time as DARTS is adopted as the inner loop optimizer. We hypothesize that the improvement over DARTS is due to the fact that *Hyperion* works in a way similar to DARTS due to its inner optimization loop but forces restart through multi-trials such that it can avoid getting stuck in a local minima.

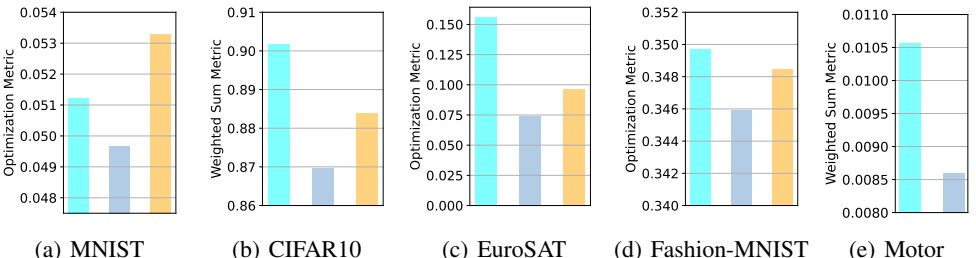

(a) MNIST (b) CIFAR10 (c) EuroSAT (d) Fashion-MNIST (e) Motor

Figure 4: Comparison of ■ BlendSearch, its fusion ■ Hyperion and ■ DARTS for different datasets.

It is important to note that DARTS is not capable of optimizing the joint hyperparameter and neural architecture search space (details in Table 1) for Motor dataset (explained in Appendix C). We further show the impact and the necessity of performing joint hyperparameter and neural architecture optimization for the industrial use case in Appendix C.

## 4.3 OPTIMIZATION REPRODUCIBILITY

We continue to investigate whether the improvement *Hyperion* brings in optimization is reproducible across different baseline multi-trial algorithms and across repeated experiment runs. To achieve this, we implemented Anneal and TPE additionally to BlendSearch as well as their extensions in *Hyperion*. We focus on MNIST and CIFAR10 datasets here; for each algorithm and dataset combination, we report the best model found for each run while repeating each experiment 10 times. We present the results in Figure 5, which finished in around 35 days on two A6000 GPUs (details shown in Appendix F). For MNIST, we only optimize model accuracy while for CIFAR10 we optimize both model accuracy and size with a weighted sum. We observe that *Hyperion* improves reliably over all three baseline multi-trial algorithms in terms of optimization metric 3.1% - 8.7%, top-1 accuracy for MNIST 0.07% - 0.17%, weighted sum metric (Equation 2) 2.3% - 3.1%, model size 13.8% - 30.4% and top-1 accuracy for CIFAR10 1.56% - 2.49%. Furthermore, we observe that *Hyperion* improves reliably over the baseline one-shot algorithm in terms of optimization metric 1.9% - 9.3%, top-1 accuracy for MNIST 0.028% - 0.12%, weighted sum metric 2.3% - 3.01%, model size 23.9% - 33.5% and top-1 accuracy for CIFAR10 0.6% - 2.06%. We note that the accuracy improvement is not big as the baseline algorithms can already find good models. However, *Hyperion* consistently improves over 10 separate experiments.

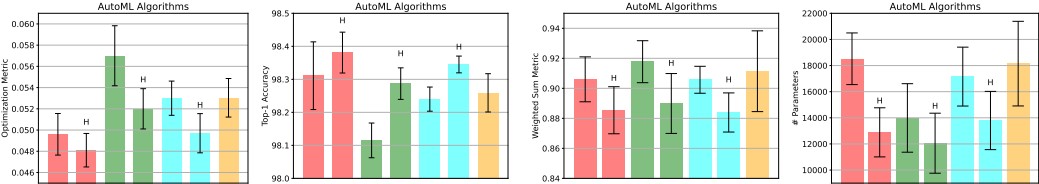

(a) MNIST opt. metric (b) MNIST top-1 accuracy (c) CIFAR10 weighted sum (d) CIFAR10 # parameters

Figure 5: Average and standard deviation of optimization metrics for MNIST and CIFAR10 datasets with ■ Anneal, ■ TPE, ■ BlendSearch, their fused versions Hyperion (marked with H) and ■ DARTS.

### 4.4 OPTIMIZATION EFFICIENCY

Standalone one-shot algorithms typically have high demand on GPU resources as they optimize a super network containing many candidate architectures. *Hyperion* overcomes this by using the smart splitter and smart sub-search-space sampler as explained in Section 3. To evaluate this, we define a search space with 5 operations, 2 activation functions and 3 kenrnel sizes (details in Table 3). We compare the GPU consumption of DARTS and *Hyperion*, see Figure 6. We observe that *Hyperion* reduces GPU resource usage by 52.17% - 61.97%, demonstrating its great resource efficiency. For *Hyperion* based on other multi-trial algorithms, similar reductions of GPU usage are observed, see Appendix G. We believe this result is important as *Hyperion* delivers better optimization performance while consuming much less resources.

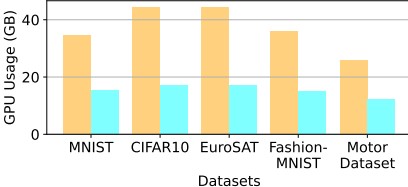

Figure 6: GPU consumptions of ■ DARTS and ■ *Hyperion* (with Anneal) across different datasets.

### 4.5 ABLATING *Hyperion*'S DESIGN

We ablate the design of *Hyperion* by presenting the learnt splitting of architecture parameters between multi-trial and one-shot optimizers, as well as the learnt sub-search-space for the one-shot optimizer. We plot how the learnt splitting and sub-search-space evolve as *Hyperion* progresses in Figure 7(a) and Figure 7(b), respectively. We select the industrial use case (motor dataset) for experiments here due to its large search space regarding both hyperparameters and neural architecture parameters. This would help us to identify emerging splitting and downsampling patterns.

*Smart Splitter.* We observe it quickly learns to discard assigning all architecture parameters to the one-shot optimizer, which would be costly to run. It learns to focus on assigning two network parameters to the one-shot optimizer, with the rest assigned to the multi-trial optimizer.

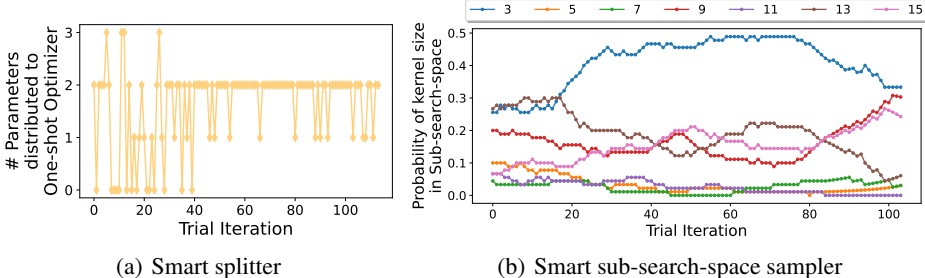

|     (a) Smart splitter     |     (b) Smart sub-search-space sampler     |

Figure 7: Number of parameters assigned to the one-shot optimizer by the smart splitter and subsampled kernel values by the smart sub-search-space sampler as *Hyperion* optimization progresses.

*Smart Sub-search-space sampler.* Our results show that *Hyperion* learns to focus on trying a converged subset of kernel sizes in the one-shot optimizer by eventually focusing on sampling subsearch-space {3,9,15} for kernel sizes, which would still contain best kernel choices.

## 5 CONCLUSION

We present *Hyperion*, which combines the strengths of multi-trial and one-shot NAS algorithms in joint hyperparameter and neural architecture optimization. *Hyperion* consists of different learning-based optimizers working in tandem to solve the joint optimization problem. We conduct extensive experiments by applying *Hyperion* to CIFAR10, MNIST, EuroSAT, Fashion-MNIST and an industrial use case on motor health prediction. Based on our evaluation, *Hyperion* requires significantly less GPU memory compared to DARTS, while achieving better optimization effectiveness compared to standalone multi-trial and one-shot algorithms.

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

## A    CHOICE OF SEARCH ALGORITHMS FOR MULTI-TRIAL OPTIMIZERS

For multi-trial optimizers in *Hyperion*, in principle, one can choose freely the underlying optimization algorithms to be integrated, for each of the involved optimizers (smart splitter, sub-search-space sampler, pipeline and architecture optimizers). To get a good coverage and to compare performances of different state-of-the-art multi-trial algorithms applied to *Hyperion*, we integrated a few typical algorithms, which allow the user to select during optimization. Specifically, *Hyperion* supports grid and random search as a baseline reference. It also supports popular Bayesian optimization based algorithms like TPE (Bergstra et al., 2011), SMAC (Hutter et al., 2011) and BOHB (Falkner et al., 2018a). Additionally, BlendSearch (Wang et al., 2021), a latest lightweight multi-trial algorithm combining both Bayesian optimization and reinforcement learning is also supported.

## B    REDUCED SEARCH SPACE FOR SUPERNET

We continue to analyze the reduction of the supernet search space in *Hyperion*. Formally, for each architecture parameter $\theta_j$ available to the $j$th node in the search space $\Theta_{arch}$, the total number of possible operations from any node to node $j$, $m_j$, equals $\prod_{\theta_j \in \Theta_{arch}} n_{\theta_j}$, where $n$ represents the number of choices available for a corresponding parameter. As each node in the supernet can have connections from all previous nodes in a DARTS cell, the total number of possible connection combinations to $j^{th}$ node, $c_j$, is then given by: $c_j = (m_j)^{j+2} = (\prod_{\theta_j \in \Theta_{arch}} n_{\theta_j})^{j+2}$. Here it is assumed that only one edge should be selected in the end between any pair of nodes. Additionally, 2 is added to the exponent to count connections from the two input nodes, as shown in Figure 1. As a result, the total number of connection combinations in a cell of the super network, $c_{cell}$, with $K$ nodes is given by $c_{cell} = \prod_{j=0}^{K-1} c_j$.

With the above analysis we can understand the super network search space reduction with a more complicated example. Assume we have the following parameters for the cell search space: kernel sizes = {3, 5, 7, 9, 11, 13, 15, 17, 19}, activations = {Relu, Tanh, Linear, Logistic, Heaviside, Gaussian}, and operations= {Conv, Dilated Convolution, Depthwise Separable Convolution, Max Pooling, Average Pooling}.

In the original DARTS, each kernel size can be combined with each activation function and layer operation to generate a candidate operation. Therefore, the total number of candidate operations between two nodes is 270 ($9 \times 6 \times 5$). This results in a large number of connection combinations in the super network. Assuming 5 nodes in the basic cell, it has ($270^2 \times 270^3 \times 270^4 \times 270^5 \times 270^6$) possible connection combinations. This shows the huge combinatorial search space and high computational demand of super networks in DARTS.

In *Hyperion*'s extension of DARTS, architecture search space $\Theta_{arch}$ is divided, such that $\Theta_{arch\_m}$ is searched by the multi-trial network optimizer and $\Theta_{arch\_o}$ is optimized by DARTS. Accordingly, $m_j$ becomes $\prod_{\theta_j \in \Theta_{arch\_o}} n_{\theta_j} \prod_{\theta_j \in \Theta_{arch\_m}} n_m$, where $n_m$ denotes the number of values the multi-trial optimizer chooses for each architecture parameter and node in the basic cell. $n_m$ can be typically set as small numbers such as 1 or 2, see Section 3.2.

For the example above, assuming that kernel sizes are distributed to the multi-trial optimizer. Based on our analysis, the total number of candidate operations between two nodes reduces significantly from 270 to 60 ($5 \times 6 \times 2$, assuming $n_m = 2$) and $c_{cell}$ reduces by a factor close to $10^{13}$. Such search space reduction greatly improves resource usage of DARTS and it does not compromise the overall search space of *Hyperion* as the multi-trial network optimizer still ensures the entire architecture search space can be explored.

## C    DETAILED RESULTS ON IMPACT OF *Hyperion* ON AN INDUSTRIAL USE CASE

We present results of applying *Hyperion* to industrial motor health prediction. The problem is to classify motor bearing status (healthy, degraded and damaged) with monitored accelerometer data. Hyperparameters like window size of the timing series taken as input, sensing frequency of the accelerometer and other training based hyperparameters like learning rate and batch size are opti-

mized together with the neural architecture by *Hyperion*, see Table 1. Additionally, the number of cells (network depth) is also optimized.

For the joint search space, *Hyperion* optimized a neural network with 192 trainable parameters achieving 99.91% accuracy on the validation set in a 12 hours experiment. Due to the small model size it can be easily implemented and run on industrial embedded platforms. Furthermore, to show the impact of joint hyperparameter and neural architecture optimization, we conducted two more experiments using *Hyperion* by fixing hyperparameters (we take the min/max values of each hyper-parameter respectively in those two experiments). Our results show that *Hyperion* returned neural networks with 724 and 5925 trainable parameters, while achieving 79.1% and 99.99% validation accuracies respectively. Therefore, the model generated when optimizing hyperparameters and neural architecture together would be more preferred in practice as it achieves high accuracy with a small model size.

| Hyperparameters | Design Choices | Category | Optimizer |
|---|---|---|---|
| Window size | [50, 100, 150, 200, 300, 400, 700] | Sensing | Multi-trial |
| Sampling Frequency | [52, 104, 208, 416, 833, 1660] | Sensing | Multi-trial |
| Learning Rate | [0.01, 0.05] | Training | Multi-trial |
| Batch Size | [256, 512, 1024, 2048] | Training | Multi-trial |
| Model Depth | [3,4,5] | Model | Multi-trial |
| Number of Channels | [1,2,3,4,5,6] | Model | Multi-trial |
| Kernel Size | [3,4,5,7,9,11,13,15] | Model | Fusion |
| Activation Function | [ReLU, Tanh] | Model | Fusion |
| Layer Operations | [Maxpool, Skip Connection, Conv2D, Identity] | Model | Fusion |

Table 1: Search space for the motor health prediction problem.

# D    DETAILED SETUP OF *Hyperion*' EXPERIMENTS

CIFAR10
$$\text{weighted\_sum} = \text{training\_loss} + 0.00001 \times \text{\#parameters} \tag{2}$$

Motor Dataset
$$\text{weighted\_sum} = (1\text{-accuracy}) + 0.4 \times (\text{\#parameters})/10000 \tag{3}$$

| Hyperparameters | Design Choices | Optimizer |
|---|---|---|
| Layer Operations | [Average Pooling, Skip Connection, Dilated Convolution] | Fusion |
| Kernel Sizes | [3, 5, 7, 9] | Fusion |
| Activation Functions | [ReLU, Tanh] | Fusion |

Table 2: Search space for showing effectiveness and reproducibility of *Hyperion*.

| Hyperparameters | Design Choices | Optimizer |
|---|---|---|
| Layer Operations | [Average Pooling, Max Pooling, Skip Connection, Dilated Convolution, Separable Convolution] | Fusion |
| Kernel Sizes | [3, 5, 7, 9] | Fusion |
| Activation Functions | [ReLU, Tanh] | Fusion |

Table 3: Search space for showing efficiency of *Hyperion*.

# E    DETAILED RESULTS ON *Hyperion*'S OPTIMIZATION EFFECTIVENESS

We observe that for all datasets *Hyperion* is able to achieve better optimization metrics compared to the the baseline multi-trial optimizer (BlendSearch) and one-shot optimizer (DARTS) during optimization as can be seen in the Figure 8. This shows the effectiveness of *Hyperion* for optimization.

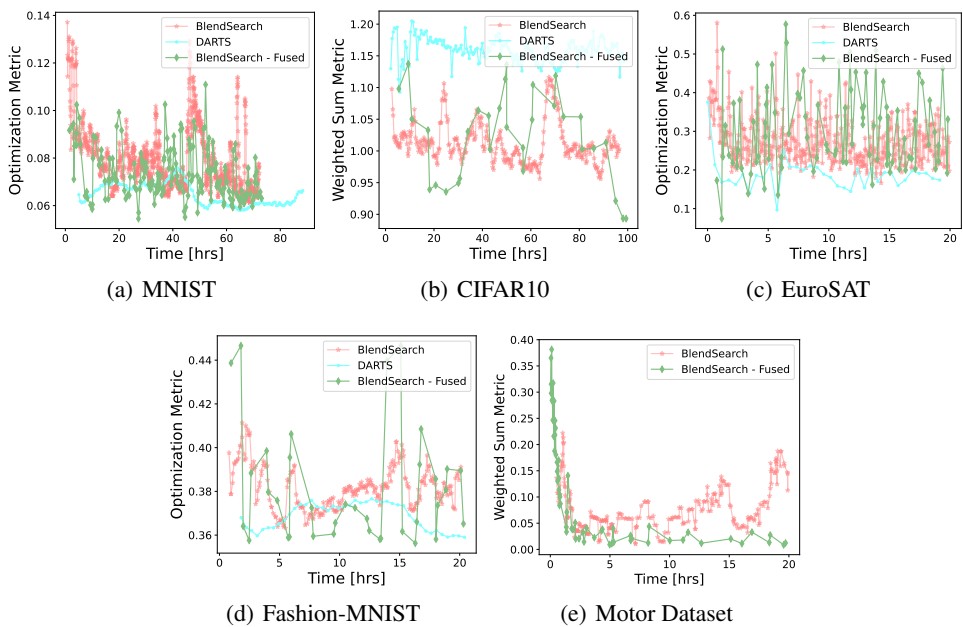

(a) MNIST    (b) CIFAR10    (c) EuroSAT

(d) Fashion-MNIST    (e) Motor Dataset

Figure 8: Comparison of moving averages of optimization targets for BlendSearch w/wo Hyperion and DARTS.

## F   DETAILED RESULTS ON *Hyperion*'S OPTIMIZATION REPRODUCIBILITY

We observe that for MNIST and CIFAR10 datasets *Hyperion* performs better optimization when averaged over 10 experiments compared to the baseline multi-trial optimizers (Anneal, TPE, Blend-Search) and one-shot optimizer (DARTS) during optimization as can be seen in the Figure 9. This shows that *Hyperion* consistently outperforms the baseline algorithms across multiple experiments and has high reproducibility.

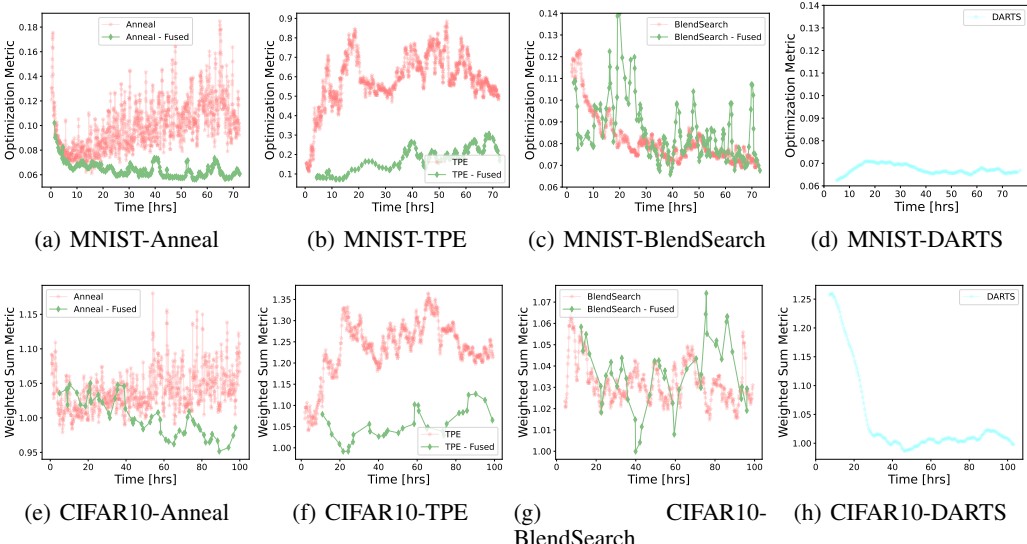

(a) MNIST-Anneal    (b) MNIST-TPE    (c) MNIST-BlendSearch    (d) MNIST-DARTS

(e) CIFAR10-Anneal    (f) CIFAR10-TPE    (g)    CIFAR10-BlendSearch    (h) CIFAR10-DARTS

Figure 9: Averaged moving averages of optimization targets across 10 complete experiments for MNIST dataset (Figures a-d) and CIFAR10 (Figures e-h) with various AutoML methods and their fused versions in Hyperion.

## G    DETAILED RESULTS ON *Hyperion*'S OPTIMIZATION EFFICIENCY

We define two search spaces for this experiment. In both search spaces, there are 5 operations and 2 activation functions. In the small search space, we have 2 design choices for the kernel size {3,5} whereas in the larger search space, we have 3 kernel choices {3,5,7}. We run DARTS on both search spaces and use this as a baseline to compare with *Hyperion*

| Search Space | AutoML Algorithms | MNIST | |
| --- | --- | --- | --- |
| | | Optimized Loss | GPU Usage |
| Large | DARTS | Not feasible | 30.21 GB |
| Small | DARTS | 0.020 | 17.34 GB |
| Large | Anneal as *Hyperion* | 0.011 | 10.5 GB |
| | **% improvement** | **45** | **39.44** |
| | TPE as *Hyperion* | 0.011 | 10 GB |
| | **% improvement** | **45** | **42.32** |
| | BlendSearch as *Hyperion* | 0.012 | 9.3 GB |
| | **% improvement** | **40** | **46.33** |

Table 4: Comparison of the optimized loss and GPU memory usage of DARTS against *Hyperion* (with different multi-trial optimizers) on the MNIST dataset.

As shown in Table 4, DARTS is only able to perform on the small search space as GPU usage for the large search space exceeds the hardware limit. *Hyperion* is able to perform on the large space as it smartly offloads certain parameters to the outer optimization loop and can further sample a sub-search-space for those parameters assigned to the inner optimization loop with DARTS. Despite using a large search space, we observe that *Hyperion* still requires less GPU memory usage than DARTS, while achieving better optimized loss metric due to its efficient exploration combining both multi-trial and one-shot optimizers.

*Other baseline multi-trial algorithms and datasets.* We observe that for the same dataset, Hyperion with different baseline multi-trial algorithms would give exact same peak GPU memory consumption during experiments. This is due to the fact that we enforce same search space. Smart splitter would assign the same max number of architecture parameters to the one-shot optimizer. Furthermore, sub-search-space sampler would sample the same sub-search-space size for the one-shot optimizer. Therefore, results would be the same as shown in Figure 6.

