# OpenReview forum: "Hyperion: Fused Multi-Trial and Gradient Descent for Joint Hyperparameter and Neural Architecture Optimization"
_ICLR.cc/2024/Conference — Submitted to ICLR 2024_

### Official Review · Reviewer_uHaA · 2023-10-28

**Soundness:** 2 fair
**Presentation:** 2 fair
**Contribution:** 2 fair
**Rating:** 1
**Confidence:** 5

**Summary:**

The paper introduces Hyperion, a framework designed to execute both hyperparameter optimization (HPO) and neural architecture search (NAS) simultaneously by fusing multi-trial and gradient descent-based one-shot algorithms, thereby aiming to provide more optimal solutions compared to treating HPO and NAS as separate tasks. The authors propose the incorporation of a smart splitter and a smart sub-search-space sampler to allocate parts of the search space and reduce overall GPU resource consumption. Combined with optimizers like TPE, along with the gradient descent-based one-shot algorithm DARTS, the authors aim to facilitate a coordinated optimization of hyperparameters and neural architectures. They execute several experiments on open image datasets and an industrial data to support their claims.

**Strengths:**

The paper addresses an important problem. In fact, multi-trial (e.g., Bayesian Optimization) and gradient-based NAS was not properly combined before and both have their strength in weaknesses. So, any approach solving this problem convincingly (i.e., getting the best of both worlds) would be a significant contribution to AutoML. In a sense, this also implies that there is quite some novelty by combining existing ideas and trying to come up with something superior.

Although there are possible improvements regarding clarity (see weaknesses), overall the approach is well motivated and Algorithm 1 explains the major idea well.

Quality is unfortunately lacking and is further discussed under weaknesses.

**Weaknesses:**

### Significance and Originality

My biggest problem is that the disadvantages of multi-trial and one-shot algorithms discussed at the beginning of the paper, e.g., instability of DARTS and many trainings of multi-trial approaches, but overall, these are retained in the Hyperion algorithm. Although Hyperion shows some strong performance, I wondered whether it actually it is not the best of both worlds but also the worst of both worlds.

### Clarity

The writing style of the paper lacks clarity. Although the main motivation is clear, a full understanding and how all the components play together was only fully understandable on Page 6. That’s too late and makes it hard to read the paper. The main problem might lie in the emphasis on gradient-based approaches, which don’t need to restart training from scratch all the time. Therefore, I expected that Hyperion would also have this property. That this is not the case, but only for the inner DARTS loop, was only clear to me on Page 6.

It also confused me that the authors jump between hyperparameters and parameters. In fact, I believe that they sometimes call hyperparameters parameters. This needs careful consideration again.

Another problem is the multi-objective statement. In fact, the authors say that they do multi-objective optimization, but in the end it is only an apriori scalarization of two objectives; this boils it down to single objective. Real multi-objective with Pareto Fronts and so on is never happening.

Further problems in clarity are shown by my rather long list of questions (see below).

### Quality of Experiments

The experiments overall are a major problem. I have several major concerns.
The experiments in Figure 4 are evaluated using varying metrics and, therefore, lack comparability. The reasoning behind this is not explained.

The datasets are rather old and small – not necessarily problematic. But for MNIST, CIFAR10 and co state of the art is very strong and well known. Because of the obscure metrics, it is unclear whether they achieved state-of-the-art on these datasets. If not, I strongly wonder about the significance of the results.

Furthermore, it is not clear whether the performance gains are significant or not. E.g., on MNIST, the improvement is 0.001. In addition, Figure 4 lacks uncertainty (e.g., standard deviations across repeated experiments); so we don’t know whether these results are by chance.

I also wondered why MNIST got 72h runtime, while FASHION-MNIST, EuroSAT and Motor only got 20h each. MNIST is the most trivial vision dataset and it is very trivial to get SOTA performance on it with much less time. In general, I would be interested to see how the approaches compare over time (as shown in nearly all AutoML papers).

There is no comparison against current state-of-the-art AutoML approaches. TPE is fairly old and outdated. I would expect at the very least a comparison against multi-fidelity Bayesian Optimization since it also partially solves the problem of training the DNN always full and thus is somehow inbetween black-box multi-trial (i.e., many full trainings) and gradient-based NAS (i.e., a single full training). This is in particular important since Auto-PyTorch (using BOHB) already showed that it outperforms multi-trial black box optimization and gradient-based DARTS.

Last but not least, the motor dataset puzzles me. I don’t know why this dataset was considered in this paper and whether this is somewhere publicly available for reproducibility reasons.

### Related Work

In fact, addressing joint HPO and NAS is not new. The novelty only comes from combining DARTS and multi-trial HPO. Previously published papers on joint HPO and NAS are not discussed in the paper and are not compared against. For example, JAHS or Auto-PyTorch.

### Minor comments

* The claim that Bayesian optimization methods learn optimized sub-search spaces is incorrect (Section 3.2).
* Despite it being mentioned in Section 4.2, there is no justification in the appendix about the motor dataset in combination with DARTS.
* I would not call Section 4.5 an ablation study. It is not ablating any component of Hyperion. Nevertheless, it is a valuable subsection.

**Questions:**

* The authors state that “invalid distributions are simply discarded” in the smart splitter in Section 3.2. What exactly does that entail?
* Why is setting up a model in Algorithm 1 called “FitModel()”? The term usually refers to training a model.
* Why are the hyperparameters defining data preprocessing, etc., in Section 4.1 fixed, and to what values? The introduction implies that tuning these would be a contribution of the paper.
* Is there a way to quantify the uncertainty in the experiments in Section 4.2? Could noise level and variance be provided? Were multiple experiments run in this setting? Are the results on CIFAR10 statistically significant?
* Are the results in Figure 5 b and c statistically significant?
* Could the authors elaborate on the scalability of Hyperion when dealing with larger datasets or more complex architectures? The runtimes for the baseline datasets seem rather long. The search spaces evaluated in the experiments are very limited in their size.
* Section 4.5 . Why does the probability of the kernel sizes in Figure 7(b) differ from the start? How does that impact the results?

---

### Official Review · Reviewer_nymt · 2023-10-30

**Soundness:** 2 fair
**Presentation:** 3 good
**Contribution:** 2 fair
**Rating:** 5
**Confidence:** 5

**Summary:**

The authors propose a method for joint hyperparameter optimization and neural architecture search. It consists of several components: a smart splitter which decides which hyperparameter will be solved by a classic HPO method and which will be optimized via DARTS. A search space sampler which selects a subset of feasible hyperparameters from the ones assigned to DARTS. A classic HPO method that selects its part of the hyperparameter settings and then the DARTS component which optimizes the remaining ones with gradient descent. All decision making components (besides DARTS), will learn over time to make better decisions.
This system is evaluated on 5 image datasets in terms of custom optimization metrics and GPU memory usage.

**Strengths:**

The authors address with joint HPO and NAS an interesting research topic. The proposed system contains of many components, but the choice seems technically solid. The system is evaluated on 5 different datasets and the authors address the GPU memory problem which is oftentimes an issue with DARTS. The ablation studies shed some light on the design choices made.
Given the complexity of the system, I think it is pretty well described.

**Weaknesses:**

Most of the weaknesses of this paper are related to the empirical evaluation.

1. The authors discuss related work on joint NAS and HPO, but do not compare to any of it.

2. There is no motivation of using other metrics than top-1 accuracy in this particular experimental setup:
    - Creating a custom metric that combines number of parameters and accuracy gives DARTS an unfair disadvantage. In that case a comparison to a method optimizing for such a metric should be considered instead [1].
    - The authors claim that the optimization metric for MNIST, Fashion-MNIST and EuroSAT is accuracy. This is clearly not the case as can be seen in Figure 4 and 5.
    - It is not possible to compare the reported results to other reported numbers out there since we in most cases do not have any idea what the top-1 accuracy is.

3. The authors claim savings in GPU memory. They do not demonstrate these benefits by, e.g., training on ImageNet, nor do they discuss other works that overcame this shortcoming of DARTS already [1].

4. Unclear whether the results are statistically significant: Figure 5 implies no significance. Can you provide the standard deviation for Figure 4?

5. Search budget seems very high for all datasets: Why is the search budget for MNIST 72 GPU hours? Training a simple 2 layer CNN will reach the same performance in just 5 minutes. What is the search time for each method? I doubt that DARTS really needs 72 hours.

Not related to the evaluation, but certainly also a small weakness is the fact that the system is rather complex. I wonder how many hyperparameters you have to set and how sensitive your system is to that.

Concluding, my main concerns are choice of baselines, metrics, statistical significance, and choice of search time. Right now, I am leaning more towards rejection, but there are many open questions that I would like the authors to address.

Thanks for your efforts!

[1] Han Cai, Ligeng Zhu, Song Han: ProxylessNAS: Direct Neural Architecture Search on Target Task and Hardware. ICLR 2019

**Questions:**

Have you tried to run a simple black box method such as BO to optimize the parameters that you cannot optimize with DARTS and basically use the DARTS component as your black box?

Can you optimize for top-1 accuracy instead or at least provide a convincing reason why you have chosen these optimization metrics?

I left couple questions more above in the weaknesses field.

---

### Official Review · Reviewer_sktS · 2023-11-01

**Soundness:** 2 fair
**Presentation:** 2 fair
**Contribution:** 2 fair
**Rating:** 3
**Confidence:** 5

**Summary:**

This paper proposes an approach to jointly optimize hyperparameters and neural network architectures efficiently. The authors advocate for the utilization of different optimizers from both blackbox and one-shot categories in an interleaving way, and introduce a mechanism to distribute (architectural and non-architectural) parameters among them. Furthermore, Hyperion utilizes a sub-space selection technique to reduce search costs associated with one-shot algorithms such as DARTS, thereby potentially mitigating the computational resources required. The authors evaluate Hyperion on simple image classification tasks and show improvements compared to baselines in both resource utilization and performance.

**Strengths:**

- The significance of the joint hyperparameter and architecture optimization is high in the AutoML community and it is definitely a very difficult problem to tackle. The blend of blackbox optimizers and one-shot algorithms is interesting and one of the straightforward ways to approach this problem based on previous research.  This fusion approach may also be relevant especially in resource-constrained scenarios.

- One of the advantages of Hyperion, as highlighted in the paper, is its ability to reduce the GPU resources needed for one-shot NAS algorithms, by conducting the gradient-based NAS search on sampled subspaces of the original space. This is a critical consideration in practical deep learning applications, where computational resources can be a limiting factor.

- The empirical evaluations are conducted in a handful of datasets and the results demonstrated that Hyperion consistently outperformed the blackbox optimization and DARTS in terms of optimized metrics with reduced search costs.

- In general, the paper is well-written and the structure of the paper allows an easy navigation of it.

**Weaknesses:**

Below I list my main concerns regarding this submission:

**Novelty and Related Work**

- Despite the difficulty and motivation of the problem at hand, I think the paper is limited in terms of novelty and contributions. Firstly, the proposed approach was also used in AutoHAS [1] (or earlier versions of the paper) and similar to it Hyperion still seems to struggle with the computational costs required for search (72 hours search on MNIST).

- The partitioning of the full search space into sub-spaces is not novel in the NAS community, even though it seems a reasonable approach to reduce the memory footprint during search in Hyperion. Some examples (that I think should also be mentioned in the related work) are LaMCTS [2], that uses Monte Carlo Tree Search to learn partitions of NAS search spaces and more closely related, Few-shot NAS [3] that splits the one-shot network to sub-one-shot networks.

- Another recent and probably the most relevant work is DHA [4], which efficiently jointly optimizes the data augmentation policy, hyperparameters and architecture.

**Empirical evaluation**

- The baselines that the authors picked in their experiments are not the strongest. Firstly, there are many one-shot NAS methods that have much lower memory footprint compared to DARTS (e.g. PC-DARTS [5]), and comparing to them should be fairer.

- The benchmarks the authors picked for their experiments are way too trivial. CIFAR-10 is of course a standard benchmark in NAS, however running on that only, and more trivial ones as MNIST, is not sufficient to claim the conclusions the authors state.

- The runtime of Hyperion seems still to be non-negligible. It requires 96 h on CIFAR-10 and 72 h on MNIST and this is way too much time in practice if a practitioner would need to run Hyperion on every new benchmark.

- A comparison to existing joint NAS and HPO algorithms such as DHA [4] and the one from Zela et al. (2018) [6] is lacking.

**Clarity**

- I found it difficult to follow the description of various algorithmic components in section 3.4, which goes along with Algorithm 1. It would be great if the authors would provide a higher-level description of the algorithm in the main paper and move the details to the appendix.

**Reproducibility**

- I wasn't able to find the code, so this hampers the reproducibility.

**References**

[1] https://arxiv.org/abs/2006.03656

[2] https://arxiv.org/pdf/2007.00708.pdf

[3] http://proceedings.mlr.press/v139/zhao21d/zhao21d.pdf

[4] https://openreview.net/pdf?id=MHOAEiTlen

[5] https://arxiv.org/abs/1907.05737

[6] https://arxiv.org/pdf/1807.06906.pdf

**Questions:**

- There exists a plethora of follow-up work based on DARTS that aim to tackle the computational costs involved with DARTS. Some notable ones that the community uses are GDAS [1], SNAS [2], PC-DARTS [3]. Therefore, I did not fully understand the choice the authors made to select DARTS as their one-shot optimizer and then incorporate the sub-space partition method to circumvent the memory issues associated with DARTS, when these methods already address these issues. Can the authors say a few words regarding this choice?

- Could you please provide more details on the costs involved with the search? Which part of the search takes most of the time? Why did the experiments in Fig. 5 took 70 GPU days?

- The authors cite the work from Zela et al. (2018) [4], however they do not compare to it in their experiments. Is there any reason why?

**References**

[1] https://arxiv.org/abs/1910.04465

[2] https://arxiv.org/pdf/1812.09926v3.pdf

[3] https://arxiv.org/abs/1907.05737

[4] https://arxiv.org/pdf/1807.06906.pdf

---

### Official Review · Reviewer_sixZ · 2023-11-03

**Soundness:** 2 fair
**Presentation:** 2 fair
**Contribution:** 2 fair
**Rating:** 3
**Confidence:** 4

**Summary:**

This paper proposes to combine multi-trial search and gradient decent to joint optimize hyper-parameters and neural architectures.  It combines the advantage of DARTS (gradient decent-based NAS) and conventional multi-trial search (such as NASNet), and expand the conventional neural architecture search to hyperparamter optimization.  The idea is intuitive, but the results are mostly limited to tiny datasets.

**Strengths:**

1. Good research problem. Researchers often optimize neural architectures and hyper-parameters separately, leading to a sub-optimal solution.  This paper aims to jointly optimize both, and can potentially lead to better results.
2. Intuitive ideas.  This paper is not the first to jointly optimize network and hyper-parameters, but in general, this is an unsolved and difficult challenge, because they naturally require different approaches to optimize. The idea proposed in this paper is to combine both multi-trial and gradient decent, which is a good direction to explore.
3.

**Weaknesses:**

1. The results are quite limited.  Most of datasets are tiny: MNIST, CIFAR, Fashion-MNIST.  Because those datasets are too small, you may have a risk of overfitting to the dataset, while jointly optimizing hyper-parameters and networks. It is also difficult to justify if the proposed method can generalize to large datasets, such as ImageNet, COCO, etc.

2. The method is a bit over complicated.  Because it combines multi-trial and gradient decent, it is somewhat expected that the method will be complicated. Looking at Algorithm 1, it has quite many steps with different optimizers, making the algorithm not easy to understand.

3. Search space is quite limited. For example, learning rate only has two values in Table 1. Not sure if the method can apply to larger search space.

**Questions:**

1. Could you add more datasets, such as ImageNet and COCO?
2. Could you explain how would you handle continuous hyper-parameters? For example, people often search for a learning rate value between 1e-3 to 1e-5, but the range is continuous rather than discrete. How would you handle that?
3. Could you try some experiments with a larger search space?

---

### Meta-Review · Area_Chair_SwP2 · 2023-12-02

**Metareview:**

The paper addresses the joint optimization of hyperparameters and neural architectures which distributes search parameters into different optimizers. The reviewers’ consensus is that the work has multiple limitations in terms of the quality of experiments, as well as its clarity. In addition, the authors did not engage with the reviewers during the rebuttal. I recommend rejecting the paper and I advise the authors to carefully consider the reviewers' comments to improve their work before the next submission.

**Justification For Why Not Higher Score:**

The work has several limitations in terms of clarity and the quality of experiments.

**Justification For Why Not Lower Score:**

N/A

---

### Decision · Program_Chairs · 2024-01-16

Reject